# TMEM176B Regulates AKT/mTOR Signaling and Tumor Growth in Triple-Negative Breast Cancer

**DOI:** 10.3390/cells10123430

**Published:** 2021-12-06

**Authors:** Chifei Kang, Ran Rostoker, Sarit Ben-Shumel, Rola Rashed, James Andrew Duty, Deniz Demircioglu, Irini M. Antoniou, Lika Isakov, Zila Shen-Orr, Jose Javier Bravo-Cordero, Nathan Kase, Math P. Cuajungco, Thomas M. Moran, Derek LeRoith, Emily Jane Gallagher

**Affiliations:** 1Division of Endocrinology, Diabetes and Bone Disease, Department of Medicine, Icahn School of Medicine at Mount Sinai, One Gustave L Levy Place, Box 1055, New York, NY 10029, USA; andrewkang2016@yahoo.com (C.K.); irmarkellaantoniou@gmail.com (I.M.A.); nathan.kase@mssm.edu (N.K.); derek.leroith@mssm.edu (D.L.); 2Diabetes and Metabolism Clinical Research Center of Excellence, Clinical Research Institute at Rambam (CRIR) and the Faculty of Medicine, Technion, Rambam Medical Center, Haifa 31096, Israel; ranrosto@gmail.com (R.R.); saritbe9@gmail.com (S.B.-S.); rola_rashed@hotmail.com (R.R.); lika.isakov@gmail.com (L.I.); Zshenorr@gmail.com (Z.S.-O.); 3Department of Microbiology, Icahn School of Medicine at Mount Sinai, New York, NY 10029, USA; andrew.duty@mssm.edu (J.A.D.); Thomas.Moran@mssm.edu (T.M.M.); 4Center of Therapeutic Antibody Development, Icahn School of Medicine at Mount Sinai, New York, NY 10029, USA; 5The Bioinformatic for Next Generation Sequencing (BiNGS) Core, Tisch Cancer Institute, Icahn School of Medicine at Mount Sinai, New York, NY 10029, USA; deniz.demircioglu@mssm.edu; 6Division of Hematology and Oncology, Department of Medicine, Icahn School of Medicine at Mount Sinai, New York, NY 10029, USA; josejavier.bravo-cordero@mssm.edu; 7Tisch Cancer Institute, Icahn School of Medicine at Mount Sinai, New York, NY 10029, USA; 8Biological Science, California State University, Fullerton, CA 92831, USA; mcuajungco@fullerton.edu

**Keywords:** TMEM176B, calcium channel, triple negative breast cancer, AKT/mTOR signaling, RNA-seq, therapeutic antibodies

## Abstract

TMEM176B is a member of the membrane spanning 4-domains (MS4) family of transmembrane proteins, and a putative ion channel that is expressed in immune cells and certain cancers. We aimed to understand the role of TMEM176B in cancer cell signaling, gene expression, cell proliferation, and migration in vitro, as well as tumor growth in vivo. We generated breast cancer cell lines with overexpressed and silenced TMEM176B, and a therapeutic antibody targeting TMEM176B. Proliferation and migration assays were performed in vitro, and tumor growth was evaluated in vivo. We performed gene expression and Western blot analyses to identify the most differentially regulated genes and signaling pathways in cells with TMEM176B overexpression and silencing. Silencing TMEM176B or inhibiting it with a therapeutic antibody impaired cell proliferation, while overexpression increased proliferation in vitro. Syngeneic and xenograft tumor studies revealed the attenuated growth of tumors with TMEM176B gene silencing compared with controls. We found that the AKT/mTOR signaling pathway was activated or repressed in cells overexpressing or silenced for TMEM176B, respectively. Overall, our results suggest that TMEM176B expression in breast cancer cells regulates key signaling pathways and genes that contribute to cancer cell growth and progression, and is a potential target for therapeutic antibodies.

## 1. Introduction

TMEM176B is a tetraspanin membrane protein that belongs to the membrane-spanning 4A (MS4A) protein family [1,2]. Although TMEM176B was described in human lung fibroblasts more than 20 years ago [3,4], the understanding of its role in cancer biology remains limited. Previous studies have described TMEM176B as an acid-sensitive cation channel, and research has primarily focused on its role in immune cell regulation [4,5,6]. Myeloid lineage immune cells express TMEM176B, in which it regulates dendritic cell maturation and antigen presentation [6,7]. Higher TMEM176B expression has been found in immature dendritic cells in allograft tolerance models and in patients with spinal cord injuries [8,9], while lower expression has been reported in mature dendritic cells. In dendritic cells, TMEM176B was reported to contribute to their suppressive function by permitting the sodium counterflux required for the acidification of endophagosomes [4]. It has additionally been reported to traffic through the Golgi apparatus in TMEM176B transfected HeLa cells, although there are conflicting results regarding its subcellular localization in different cell types [4,5,6,10]. The whole-body deletion or pharmacological inhibition of TMEM176B in murine cancer models decreased tumor growth through enhanced anti-tumor T cell immunity and improved the response to immune checkpoint inhibitors [4,10]. Although the importance of TMEM176B in immune regulation is emerging [4,10], much remains to be understood about its role in intracellular processes.

Phylogenetic analysis has proposed that TMEM176 genes first appeared in cartilaginous fish, and were expressed in nonimmune cells, prior to expression expanding to immune cells in mammalian species [11]. A small number of human and rodent studies have examined TMEM176B in nonimmune cells [12,13]. Importantly, an altered expression of TMEM176B has been found in a number of cancer types, while abnormal methylation of the CpG islands associated with TMEM176B was reported in breast cancers [12,14]. The chromosome 7q36.1-3 region within which lies the gene for TMEM176B exhibits frequent gain/amplification in a number of human cancers [15]. Decreased overall survival in gastric cancer was found to correlate with higher levels of *TMEM176B* mRNA [16]. In contrast, the overexpression of TMEM176B led to decreased proliferation of the androgen-sensitive LNCaP prostate cancer cell line and reduced the growth of NIH3T3 cells transfected with constitutively active H-Ras [17,18]. The regulation of endosomal pH by cation channels in cancer cells, and their importance in cancer cell signaling, is an emerging field [19]. Overall, much remains to be understood regarding the role of TMEM176B in cancer biology.

Our interest in TMEM176B began when we identified it as the most upregulated gene in a c-Myc/VEGFA-expressing murine breast cancer cell line (Mvt1) sorted by flow cytometry based on the positive expression of the sialoglycoprotein CD24 [20]. We found that orthotopic tumors derived from the CD24-positive (CD24^+^) subset grew more rapidly than the CD24-negative cells [20]. CD24 has recently been identified as a putative oncogene, a marker of resistance to chemotherapy and a “don’t eat me” signal in ovarian cancer and triple-negative breast cancer (TNBC) [21,22,23]. In this study, we aimed to develop a greater understanding of the role of TMEM176B in TNBC cell processes.

## 2. Methods

### 2.1. Expression and Survival Studies in Publicly Available Datasets

We used cBioportal for Cancer Genomics to examine *TMEM176B* copy number amplification in breast cancer subtypes in the METABRIC dataset [24,25,26,27]. The analysis of gene expression subtype in the TCGA dataset was performed using the UALCAN cancer database [28]. Gene expression by breast cancer grade was examined using the Gene Expression database of the Normal and Tumor Tissues 2 (GENT2) dataset [29]. We used Kaplan–Meier plotter to examine the relapse-free survival according to the *TMEM176B* low and high mRNA expression from individual breast cancer studies within the dataset [30]. Detailed information of these studies can be found in the Gene Expression Omnibus (GEO), National Center for Biotechnology Information (NCBI).

### 2.2. Cell Lines

Murine and human TNBC cell lines were used in these experiments. The Mvt1, Met-1, and M-wnt cell lines were established as previously described [31,32,33]. The human TNBC MDA-MB-231 cells were validated by IDEXX BioAnalytics (Columbia, MO, USA) and were negative for interspecies contamination. All cell lines were grown in DMEM with 10% fetal bovine serum (FBS) and 1% penicillin/streptomycin (P/S), with the exception of the M-wnt cells that were grown in RPMI with 10% FBS and 1% P/S. They were all cultured in a humidified 37 °C incubator, with 5% CO_2_.

### 2.3. Generation of Stable Knockdown and Overexpression Cell Lines

Lentiviral vectors encoding microRNA-adapted short hairpin RNAs (shRNA) for *TMEM176A* and *TMEM176B* silencing and control plasmids were purchased from Genecopoeia (Rockville, MD, USA). Lentiviral vectors encoding human *TMEM176A* and *TMEM176B* and the control vector were purchased from Origene (Rockville, MD, USA). Vector transduction was performed as previously described [34]. Vectors were packaged into lentiviral particles using the viral power packaging system (Invitrogen, Burlington, ON, Canada). CD24+ Mvt-1 cells, Met-1, M-wnt, and MDA-MB-231 cells were infected in the presence of 8 μg/mL of polybrene (Sigma-Aldrich, St. Louis, MO, USA). The stable knockdown of TMEM176A and TMEM176B in MDA-MB-231, CD24+ Mvt-1 cells, Met-1, and M-wnt cells was achieved by selection with 4 μg/mL of puromycin (Sigma-Aldrich). The stable overexpression of TMEM176A and TMEM176B in MDA-MB-231 was achieved by selection with 2 mg/mL of G418 (Sigma-Aldrich). Plasmid details are shown in Appendix A.

### 2.4. In Vivo Tumor Studies

Animal studies using Mvt1 cells were performed at the Technion, Haifa, Israel, according to the protocol approved by the Technion Animal Inspection Committee. The Technion holds an NIH animal approval license number A5026-01. The in vivo studies using MDA-MB-231 human cancer cells were performed at the ISMMS, New York, NY, USA, using procedures in compliance with the current standards specified in the Guide of the Care and Use of Laboratory Animals provided by the Association for Assessment and Accreditation of Laboratory Animal Care (AAALAC) and approved by the Icahn School of Medicine at Mount Sinai (ISMMS) Institutional Animal Care and Use Committee. All mice used in these studies were female on an FVB/N background. Wild-type (WT) females were used for the Mvt1 studies, and recombination activating gene 1 knockout (Rag1^−/−^) female mice on an FVB/N background were used for MDA-MB-231 xenograft studies.

Mice were housed 4–5 per cage, maintained on a 12 h light/dark cycle, and fed a regular chow diet (PicoLab 5053, Brentwood, MO, USA). Numbers of 5 × 10^4^ Mvt1 cells/mouse and 5 x 10^6^ MDA-MB-231 cells/mouse were injected into the 4th mammary fat pad in sterile PBS. The primary outcome of this study was tumor growth assessed by tumor volumes and tumor weights. Growth was quantified using calipers, and tumor volumes were calculated using the formula: volume = 4/3 × π × (length/2 × width/2 × depth/2) [35]. Sample size and detailed tumor information in each study are shown in Appendix A. The sample size was determined from the number of mice required in our previous experiments to see differences in tumor growth between groups [36]. Mice were randomly allocated into each group in the studies. All mice were euthanized on the same day and tumor weights were measured at the time of dissection. Studies were not blinded. Body weight and body condition were monitored weekly. Mice were euthanized by cervical dislocation under anesthesia with isoflurane, and dissected once humane endpoints were reached. This animal study protocol was not registered before the study.

### 2.5. Quantitative PCR Reaction for Gene Expression

RNA isolation, reverse transcription, and quantitative PCR were performed as previously described [20,37]. Primer sequences are shown in Appendix A.

### 2.6. Western Blot Analysis

Protein isolation and Western blot analyses were performed as previously described [35]. Antibody information is shown in Appendix A.

### 2.7. Immunocytochemistry and Confocal Microscopy

TMEM176B-OE, TMEM176B-KD, and vector control MDA-MB-231 cells were plated on glass coverslips. Cells were fixed with 4% paraformaldehyde for 10 min, washed with PBS, permeabilized with 0.5% Triton X-100 (Sigma-Aldrich, St. Louis, MO, USA), blocked with 10% goat serum (Fisher Scientific, Pittsburg, PA, USA), 0.1% Triton-X100, and 10 mg/mL of BSA for 1 h, and then incubated with primary antibodies, as indicated. Details of the primary and secondary antibodies are shown in Appendix A. Cells were mounted with ProLong^®^ Gold Antifade Reagent with DAPI. Samples were imaged in the Icahn School of Medicine at Mount Sinai (ISMMS) Microscopy Core Facility using a Leica SP5 DMI confocal microscope with a 63× objective. The fluorescence intensity, normalized to cell area, was quantified using ImageJ software (NIH, Bethesda, MD, USA).

### 2.8. mRNA-Sequencing and Data Analyses

Total RNA was isolated and purified from cells using the RNeasy Mini Kit (Qiagen, Germantown, MD, USA) according to the manufacturer’s instructions. RNA-seq experiments and related quality control analyses were completed by GENEWIZ, Inc. (South Plainfield, NJ, USA). An amount of 500 ng of total RNA was taken using the NEBNext Ultra RNA library preparation kit (New England Biolabs, Ipswich, MA) to prepare cDNA libraries for each sample. Briefly, polyadenylated RNA was purified using magnetic beads and fragmented according to the manufacturer’s instructions. After ligation of the paired-end adapter, the approximately 400-bp fraction was amplified with 9 cycles of PCR. Then, cDNA libraries were subjected to the Illumina HiSeq X™ Series platform (Illumina, San Diego, CA, USA) using the 2 × 150 bp paired end (PE) sequencing configuration. Quality control of the raw reads was performed using FastQC v.0.11.8 (http://www.bioinformatics.babraham.ac.uk/projects/fastqc, accessed on 13 October 2021). Trim Galore! v.0.6.5 (https://www.bioinformatics.babraham.ac.uk/projects/trim_galore (last accessed on 13 October 2021)) was used to perform adapter and quality trimming with a quality threshold of 20. The human genome reference used was GRCh38.p13, and GENCODE release 36 was used as the transcriptome reference [38]. The alignment was performed using STAR aligner v.2.7.5b (https://github.com/alexdobin/STAR/releases (last accessed on 13 October 2021) [39].

Gene level read counts were obtained by using Salmon v.1.2.1 (https://github.com/COMBINE-lab/Salmon (last accessed on 13 October 2021) for all libraries [40]. All samples passed the quality control requirements with >90% of reads uniquely mapping (>20 M uniquely mapped reads for each library) using STAR aligner.

The gene level read counts table was used for downstream differential expression analysis. Using DESeq2 v.1.28.1 (R software package, http://bioconductor.org/packages/release/bioc/html/DESeq2.html (last accessed on 13 October 2021), pairwise comparisons of gene expression between the defined groups of samples were performed [41]. Genes with less than 5 reads in total across all samples were filtered as inactive genes. The Wald test was used to generate *p*-values and log2 fold changes (lfc). Genes with an adjusted *p*-value < 0.05 and absolute (lfc) ≥ 1 (fold-change of at least 2) were considered to be differentially expressed genes for each comparison.

The over-representation and gene set enrichment analysis for functional enrichment were both performed using the clusterProfiler v.3.16.0 (R software package, http://bioconductor.org/packages/release/bioc/html/clusterProfiler.html (last accessed on 13 October 2021) [42]. The gene sets used for functional analysis were obtained from The Molecular Signatures Database (MSigDB) [43,44,45]. The Fisher test was used to determine whether the overlap between the DEGs and the genes in the term is statistically significant (*p*-adjusted < 0.05). The bold terms with an asterisk in front are the terms that were significantly enriched (*p*-adjusted < 0.05).

We performed the between-sample normalization using the variance stabilizing transformation of the DESeq2 package. Gene expression heatmaps show the z-scores of DESeq2 VST normalized gene-level read counts. The heatmaps were generated using heatmaply v.1.1.0 (R software package, https://cran.r-project.org/web/packages/heatmaply/index.html (last accessed on 13 October 2021) [46]. All other visualizations were generated using plotly v.4.9.2.1 (R software package, https://plotly-r.com (last accessed on 13 October 2021) [47].

### 2.9. In Vitro Cell Proliferation, Migration, and Invasion Assays

Cell proliferation, migration, and invasion assays were performed as previously described [34,36]. For assessment of antibody effects on cell proliferation, cells were seeded in 24-well plates (7000 cells/well). Control mouse serum or anti-TMEM176B antibody was added into the cell medium (1:1000 dilution) 24 h after cell seeding. Control mouse serum or anti-TMEM176B antibody was refreshed daily. After 120 h, cells were counted by a hemocytometer after 1:1 dilution in trypan blue.

## 3. TMEM176B Antibody Development

To develop polyclonal antibody sera against TMEM176B, DNA representing the full-length coding sequence of TMEM176B was cloned into a pcDNA3.1 mammalian expression vector (Thermo Scientific, Waltham, MA, USA). Balb/c mice were then immunized intramuscularly with 100 µg of full-length TMEM176B DNA using electroporation. Sera were collected after three immunizations. To screen the sera for TMEM176B reactivity, the amino acid sequence for the large extracellular loop #2 (also known as the “large loop”) of TMEM176B was isolated and cloned into a Mouse IgG2a Fc fusion vector (pFuse_mIgG2aFc_2, InVivoGen, San Diego, CA, USA) with an IL-2 signal sequence at the Center for Therapeutic Antibody Discovery at the ISMMS. Expression of the TMEM176B Large Loop-Fc fusion was performed in Expi293F cells and purified using protein A HiTrap columns on an AKTA Pure Chromatography FPLC system (GE Life Sciences, Pittsburgh, PA). To test reactivity, 5 μg/mL of protein was used to coat plates for a sera enzyme-linked immunosorbance assay (ELISA). ELISAs were developed with anti-mouse kappa HRP (Horseradish Peroxidase) secondary (Southern Biotech, Birmingham, AL, USA) due to the presence of the MsIgG2a Fc fusion domain.

### 3.1. Statistical Analysis

All data are presented as mean ± SEM (standard error of the mean). Student’s *t*-tests were used for two groups of equal variance and one-way analyses of variance (ANOVAs) for more than two groups, followed by the Bonferroni multiple comparison post hoc test to determine the statistical significance of differences between groups. For RNAseq analysis, the statistical analysis is described in the “mRNA sequencing and data analyses” section. Statistical analyses were performed using Prism 8 (GraphPad Software, San Diego, CA, USA).

### 3.2. Results

#### 3.2.1. TMEM176B Expression Is Amplified in Basal-Like Breast Cancers More than Other Breast Cancer Subtypes

To evaluate the role of TMEM176B in human breast cancer, we assessed the expression of *TMEM176B* in different breast cancer subtypes in the Molecular Taxonomy of Breast Cancer International Consortium [27] and The Cancer Genome Atlas (TCGA) datasets. We found *TMEM176B* amplified in 6.5%, showing a copy number gain in 20.1% of basal-like breast cancers, which have significant cross-over with TNBC compared with other breast cancer subtypes in the METABRIC dataset (Appendix A). Expression was notably higher in the immunomodulatory subtype of TNBC compared with other subtypes in the TCGA breast cancer dataset.

#### 3.2.2. Silencing TMEM176B Inhibited Cancer Cell Proliferation and Migration In Vitro

To understand the role of TMEM176B in tumor growth, we silenced *TMEM176B* in the human basal-like MDA-MB-231 breast cancer cell line using two shRNA constructs (176Bsh1 and 176Bsh2) and confirmed the gene silencing by RNA and protein analysis (Figure 1A–C). TMEM176B silencing reduced the cell number to approximately 50% of control shRNA (Scrb) at 72 h in proliferation assays (Figure 1D). MDA-MB-231 176Bsh1 and 176Bsh2 migrated to cover significantly less area than control (Scrb) cells at 48 h (Scrb: 85%, 176Bsh1: 50%, 176Bsh2: 35%) in wound healing assays (Figure 1E,F). In transwell migration assays, 176Bsh1 and 176Bsh2 cells had 48% and 47% fewer stained cells than controls (Figure 1G,H). We further repeated these experiments in three murine breast cancer cell lines: Mvt1, Met-1, and M-wnt cells, using two *Tmem176b* shRNA constructs (176bsh1 and 176bsh2). Similar to the MDA-MB-231 cells, *Tmem176b* silencing in these cell lines also significantly impaired cell proliferation after 72 h, and inhibited migration in wound healing and transwell migration assays (Appendix A).

#### 3.2.3. Silencing TMEM176B but Not TMEM176A in Breast Cancer Cells Inhibited Tumor Growth In Vivo

To determine if silencing TMEM176B affected the growth of human breast cancer xenografts, we injected control and TMEM176B-silenced cells into the fourth mammary fat pad of immunodeficient female Rag1^−/−^ mice. Growth was significantly impaired in TMEM176Bsh1 and TMEM176Bsh2 (Figure 2).

TMEM176A and TMEM176B are closely related members of the MS4 transmembrane protein family, and have previously been found to physically interact [12], and their mRNA expression is highly correlated in the TCGA (R^2^ = 0.88) and METABRIC (R^2^ = 0.78) human breast cancer datasets (Appendix A). To understand if TMEM176A was also important for tumor growth, we silenced TMEM176A using two shRNA constructs (TMEM176Ash1 and TMEM176Ash2) in MDA-MB-231 cells (Figure 3A–C). In contrast to TMEM176B silencing, silencing TMEM176A had no effect on the growth of MDA-MB-231 xenografts (Figure 3D,E). We then generated MDA-MB-231 cells stably overexpressing TMEM176A, and TMEM176B and control cells with empty vectors (Figure 3F–H). Overexpressing TMEM176B in MDA-MB-231 breast cancer cells increased proliferation 1.5-fold in vitro, but overexpressing TMEM176A had no effect on proliferation (Figure 3I). These results strongly suggest that TMEM176B but not TMEM176A contributes to TNBC growth.

#### 3.2.4. Antibody Targeting of TMEM176B Reduced the Proliferation of MDA-MB-231 Breast Cancer Cells

We next aimed to determine if TMEM176B could be therapeutically targeted using an anti-TMEM176B antibody. We performed immunofluorescence and found that TMEM176B was located on the plasma membrane of MDA-MB-231 cells (Figure 4A). In the MS4 family of proteins, amino acid diversity is seen in the large second loop region (Figure 4B), suggesting a possible impact on functional characteristics of the receptor, and may also provide unique epitopes for making anti-TMEM176B-specific antibodies. We generated a TMEM176B polyclonal antibody (anti-TMEM176B pAb-2573), as described in the Materials and Methods section. The specificity of the antibody was verified by a sera ELISA assay (Figure 4C). We performed a proliferation assay using the anti-TMEM176B pAb-2573 (#2573) in the control, TMEM176B-overexpressing, and TMEM176B-silenced MDA-MB-231 cells. The proliferation of MDA-MB-231 in TMEM176B-overexpressing, but not TMEM176B-silenced, cells was suppressed by the anti-TMEM176B pAb-2573 (Figure 4D), suggesting that anti-TMEM176B antibodies could have therapeutic anti-cancer effects.

#### 3.2.5. RNA-Seq Revealed That TMEM176B Expression Was Associated with Differential Expression of Genes Involved in Cell Signaling

In order to understand the function of TMEM176B in TNBC cells, we performed RNA-seq analysis on MDA-MB-231 cells overexpressing TMEM176B, the two MDA-MB-231 cell lines with TMEM176B silencing, as well as their respective control cell lines. We identified 413 significantly upregulated and 155 downregulated genes (adjust *p*-value < 0.05, fold-change > 2) between the TMEM176B-overexpressing MDA-MB-231 and the control cells. We also identified 997 differentially expressed genes (DEGs) between TMEM176Bsh1 vs. control shRNA, and 191 DEGs between TMEM176Bsh2 vs. control shRNA comparisons, respectively. All differentially expressed genes identified across different comparisons are shown in the heatmap, and the differential expression status of each gene in each comparison is indicated on the sidebars (Figure 5A). From these differentially expressed genes, we looked for genes that were reciprocally regulated in the overexpression and silenced cell lines. Six genes (*MAGEA6*, *VAT1L*, *GABRA3*, *RHOXF1-AS1*, *PBX1*, *NCAM2*) were increased in TMEM176B-overexpressing cells and also decreased in both shRNA-silenced cell lines, compared with their respective control cells. Three genes (*TMEM98*, *EFHD1*, *GALNT13*) were decreased in TMEM176B-overexpressing cells and also increased in both shRNA cell lines compared with controls (Figure 5B,C). We then validated these DEGs by qRT-PCR (Appendix A).

To identify functionally enriched terms, over-representation analysis was performed using hallmark and Gene Ontology gene sets from MSigDB for all three comparisons (TMEM176B overexpression vs. control, TMEM176Bsh1 vs. shRNA control, and TMEM176Bsh2 vs. shRNA control). In the hallmark gene sets, genes involved in angiogenesis, KRAS signaling, estrogen response, the epithelial-to-mesenchymal transition (EMT), and interferon response were differentially regulated in all comparison groups (Figure 5D). Other gene sets that were differentially regulated in the GO Cellular Component and GO Biological Process ontologies are shown in Appendix A. The result of the functional analysis suggested that several intracellular functions that might be involved in the cell proliferation, migration, and tumor growth are regulated by TMEM176B.

#### 3.2.6. TMEM176B Overexpression and Silencing Impacted Activation of the AKT/mTOR Signaling Pathway

Previous studies have found that endosomes are critical hubs in cancer cell signaling, including the AKT/mechanistic target of the rapamycin (mTOR) pathway [48,49]. Ion channels have also been reported to regulate the pH of endosomes affecting cell signaling [4,19]. Studies have also found that the AKT/mTOR signaling pathway impacted the expression of a number of the genes that were differentially regulated in our cell lines (e.g., *MAGEA6, GABRA3*, *PBX1, EFHD1,* and *TMEM98),* in addition to genes involved in angiogenesis, KRAS signaling, EMT, and interferon response [50,51,52,53]. We therefore assessed the activation of the AKT/mTOR signaling pathway in TMEM176B overexpressing and silenced MDA-MB-231 cells. Phosphorylation of AKT (Thr^308^) was increased in the TMEM176B-overexpressing cells compared with controls, along with the phosphorylation of 90-kDa ribosomal S6 kinase (p90RSK) (Figure 6).

## 4. Discussion

In this study, we aimed to understand the function of the proposed ion channel TMEM176B in breast cancer. To this end, we studied the effect of TMEM176B overexpression and silencing in vitro and in vivo. We found that TMEM176B expression on breast cancer cells was important for cell proliferation and migration, AKT/mTOR signaling, and the regulation of a number of genes involved in angiogenesis, KRAS signaling, EMT, and estrogen and interferon responses. We also developed a therapeutic antibody to TMEM176B that inhibited cell proliferation. Overall, our results suggest that TMEM176B not only regulates the tumor immune microenvironment, as has been previously reported, but also directly affects cancer cells.

Previously reported to be an ion channel on endo-phagosomes that regulates their pH, the main focus of research on TMEM176B has related to its expression on immune cells and its immunoregulatory effects [4,6,7,10,12]. Studies examining the importance of TMEM176B expression in cancer cells have been limited, and reports on its intracellular localization in different cells have been conflicting [4,9]. However, the importance of ion channels and endosome pH on cancer cell signaling and phenotype is emerging [19]. In two other studies, inhibiting TMEM176B in tumor-associated cells was found to reduce tumor progression. One study found that silencing TMEM176B on melanoma-associated endothelial cells reduced their migration in vitro [13]. Indeed, in our gene expression analysis, TMEM176B overexpression and silencing affected genes involved in angiogenesis. Genetic or pharmacological inhibition of Tmem176B in the immune microenvironment improved survival in three syngeneic murine cancer models: MC38 (colon), LL/2 (lung), and EG7 (thymic lymphoma). Within those tumors, TMEM176B knockout mice had a decreased abundance of immunosuppressive regulatory T cell molecules, and a higher percentage of total and tumor-specific CD8^+^ T cells compared with control mice. Furthermore, TMEM176B knockout mice had a better response to anti-cytotoxic T-lymphocyte-associated protein 4 (CTLA-4) monoclonal antibody (mAb) treatment [10]. In our current study, we found changes in the expression of cancer cell interferon response genes, which are known to regulate T cell function [54]. Our results are reminiscent of studies on other immune checkpoints such as programmed cell death 1 (PD-1) and its ligand PD-L1. PD-1 and PD-L1 are expressed not only on immune cells but also by some tumor cells, and PD-1 expression on melanoma cells was found to activate the mTOR signaling pathway [55,56]. Interestingly, in the androgen-sensitive LNCaP prostate cancer cell line and NIH3T3 cells transfected with constitutively active H-Ras, overexpression of TMEM176B decreased cell proliferation and growth [17,18]. Previous studies in MDA-MB-231 cells have reported that inhibiting the Ras/ERK pathway did not impair cell motility and, in fact, led to a paradoxical increase in AKT/mTOR signaling [57]. These differences in response to TMEM176B highlight the complex heterogeneity between tumors and the need to further study the role of TMEM176B in the biology of different types of cancer.

We identified nine genes that were consistently regulated by TMEM176B overexpression and silencing. Three of the upregulated genes have been previously linked to AKT/mTOR signaling. *MAGEA6* encodes MAGE (melanoma-associated antigen) family member A6, and it is a ubiquitin ligase of AMP-activated protein kinase (AMPK) that is involved in regulating a number of cell processes, including endosomal protein recycling [58]. *MAGEA6* silencing was found to inhibit human colorectal and renal cell carcinoma and prevented mTOR signaling [50,59]. *GABRA3* (GABA receptor alpha3), normally exclusively expressed in the adult brain but also expressed in breast cancer, is reported to mediate AKT activation and promote breast cancer cell migration, invasion, and metastasis [60]. *PBX1* belongs to a family of pre-B cell leukemia transcription factors and is suggested to act as a pioneer factor in breast cancer, remodeling the chromatin to favor the recruitment of estrogen receptor alpha [61]. It has also been reported to activate AKT signaling [62]. The AKT/mTOR signaling pathway is known to be an important mediator of cancer progression, and its activation is associated with unfavorable outcomes [63,64]. As AKT signaling can be influenced by protein recycling, which is regulated by endosomes, and as TMEM176B is a putative cation channel that has been reported to regulate the pH of endosomes, it is possible that silencing TMEM176B alters cell signaling by affecting protein recycling [4,6,19]. However, it remains to be determined whether our observed effects of TMEM176B overexpression and silencing on AKT/mTOR signaling were directly related to TMEM176B expression, or indirectly through the altered expression of *MAGEA6, PBX1*, or *GABRA3*. When examining the survival data in the Kaplan–Meier plots (Appendix A), it is interesting to note that TMEM176B was a marker of poor prognosis in tamoxifen-treated tumors and those that received no systemic therapy. The AKT/mTOR pathway is over-activated in many ER-positive tumors and could be suppressed by tamoxifen treatment [65]. However, in the high-TMEM176B-expression cancer cells AKT/mTOR pathway, activation could be compensated even under tamoxifen suppression.

## 5. Conclusions

Overall, our results suggest that the putative ion channel TMEM176B is expressed in cancer cells, and it has important roles in regulating gene expression, as well as cell signaling. Taken together with previously published studies, the research to date suggests that TMEM176B may be important for tumor growth, potentially by direct effects on regulating the pH of endosomes, affecting tumor cell signaling and gene expression, as well as effects in the tumor microenvironment such as angiogenesis, and immune response. Targeting TMEM176B may not only enhance the anti-tumor immune response to immune checkpoint inhibitors, but may also directly inhibit tumor cell growth.

## Figures and Tables

**Figure 1 cells-10-03430-f001:**
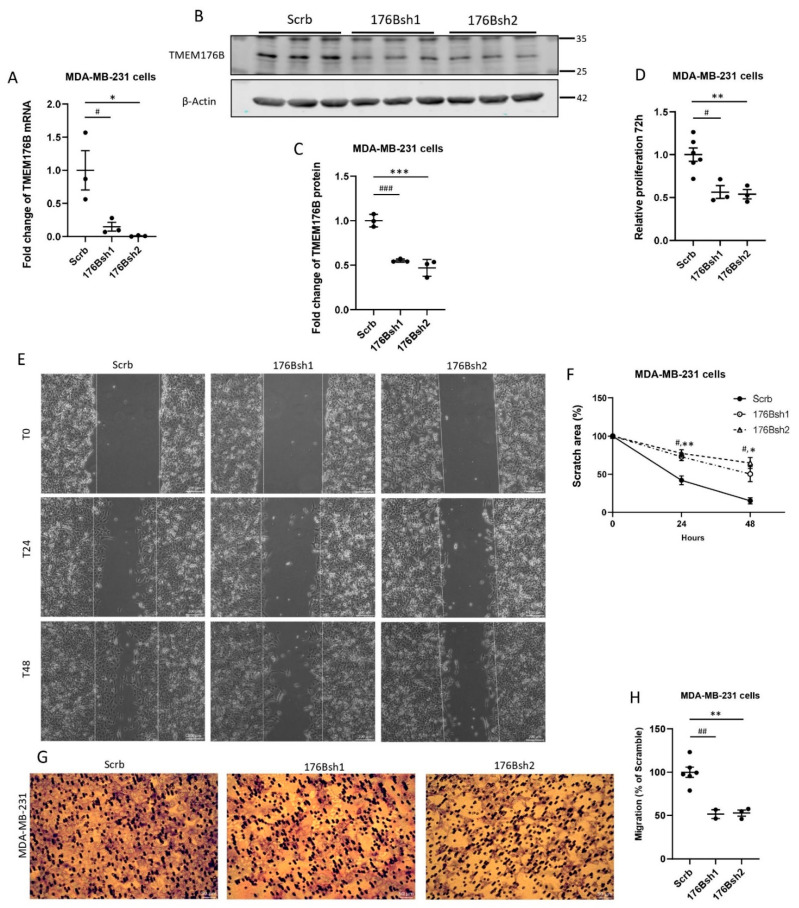
TMEM176B knockdown decreased the proliferation, wound healing, and the migration of MDA-MB-231 cells in vitro. (**A**) *TMEM176B* mRNA expression was assessed by qRT-PCR in control (Scrb) and TMEM176B-silenced (176Bsh1, 176Bsh2) MDA-MB-231 cells. (n = 3 per group, with two independent experiments). (**B**,**C**) Western blot analysis examining TMEM176B expression in control and TMEM176B-silenced cells (n = 3 per group, with three independent experiments). (**D**) Results of proliferation assay for control and TMEM176B-silenced cells grown for 72 h (n = 6 for Scrb; n = 3 for 176Bsh1 and 176Bsh2 per group, with three independent experiments). (**E**) Representative images of wound healing assay. Images were taken at time points T0 (0 h), T24 (24 h), and T48 (48 h) after performing the scratch at the same coordinates for each image. Scale bars: 200 μm. (**F**) Quantification of wound healing assay. “Scratch area %” indicates the percent of area remaining compared with time 0. (n = 3 per group, with two independent experiments). (**G**) Representative images of transwell migration assay after 20 h, stained with Giemsa solution. Scale bars, 50 μm. (**H**) Quantification of transwell migration assay. Stained area is expressed as a percent of control (Scrb) cells. (n = 6 for Scrb; n = 2 for 176Bsh1; n = 3 for 176Bsh2 per group, with two independent experiments). Data are presented as means ± SEM. Differences between groups were evaluated by the one-way (**A**,**C**,**D**,**F**,**H**) ANOVA test with the Bonferroni post hoc test. (# *p* < 0.05, ## *p* < 0.01, ### *p* < 0.001 between Scrb vs. 176Bsh1; * *p* < 0.05, ** *p* < 0.01, *** between Scrb vs. 176Bsh2) dataset (Appendix A) [28], and in histological grade 3 breast cancer compared with lower-grade breast cancers (Appendix A) [29]. High *TMEM176B* mRNA expression was also associated with decreased relapse-free survival in a majority of breast cancer studies examined (Appendix A) [30].

**Figure 2 cells-10-03430-f002:**
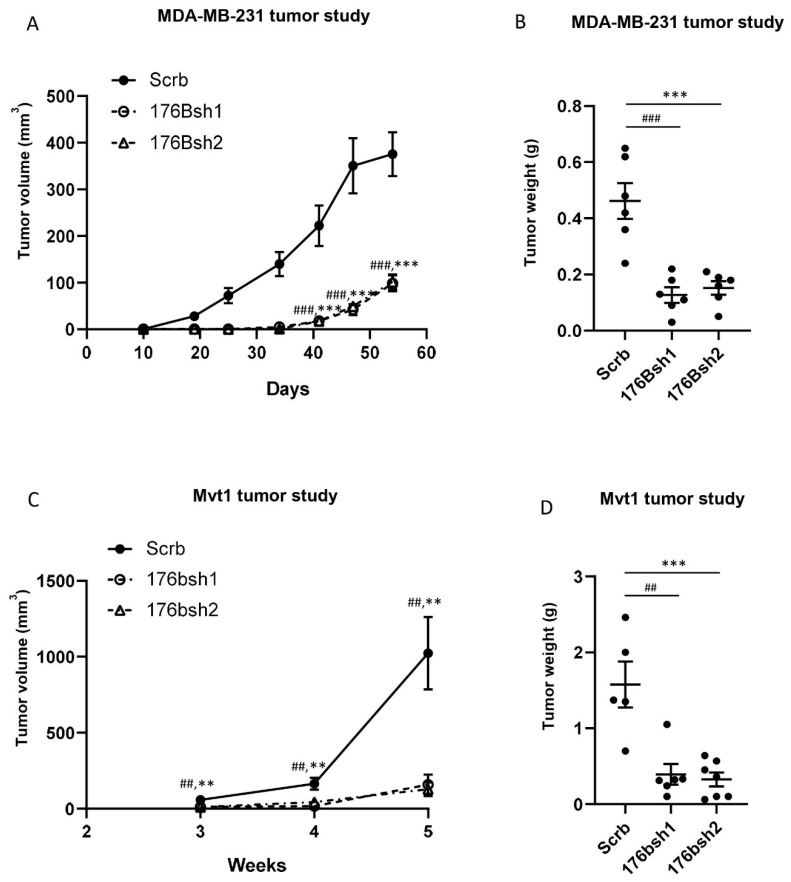
TMEM176B knockdown suppressed MDA-MB-231 and Mvt1 tumor growth in vivo. (**A**) Growth charts of MDA-MB-231 control (Scrb), and TMEM176B-silenced (176Bsh1 and 176Bsh2) tumor xenografts in the Rag1^−/−^ female mice (n = 6 mice per group). (**B**) Tumor weight at the end of the study. (**C**) Growth charts of Mvt1 control (Scrb), and Tmem176B-silenced (176bsh1 and 176bsh2) syngeneic tumors in the FVB/n female mice (n = 5–7 mice per group). (**D**) Tumor weight at the end of the study. Data are presented as means ± SEM. Differences between groups were evaluated by the one-way ANOVA test with the Bonferroni post-hoc test. ** or ## *p* < 0.01, *** or ### *p* < 0.001. (Scrb vs. 176bsh1 #; Scrb vs. 176bsh2 *) tumors compared with controls (Figure 2A,B). We next examined the growth of Mvt1 tumors with TMEM176B silencing in FVB/N mice, and we found that TMEM176B-silenced tumors displayed more than an 80% reduction in growth compared to the Mvt1 control tumors (Figure 2C,D). These findings suggest that TMEM176B expression on breast cancer cells may play an important role in tumor growth.

**Figure 3 cells-10-03430-f003:**
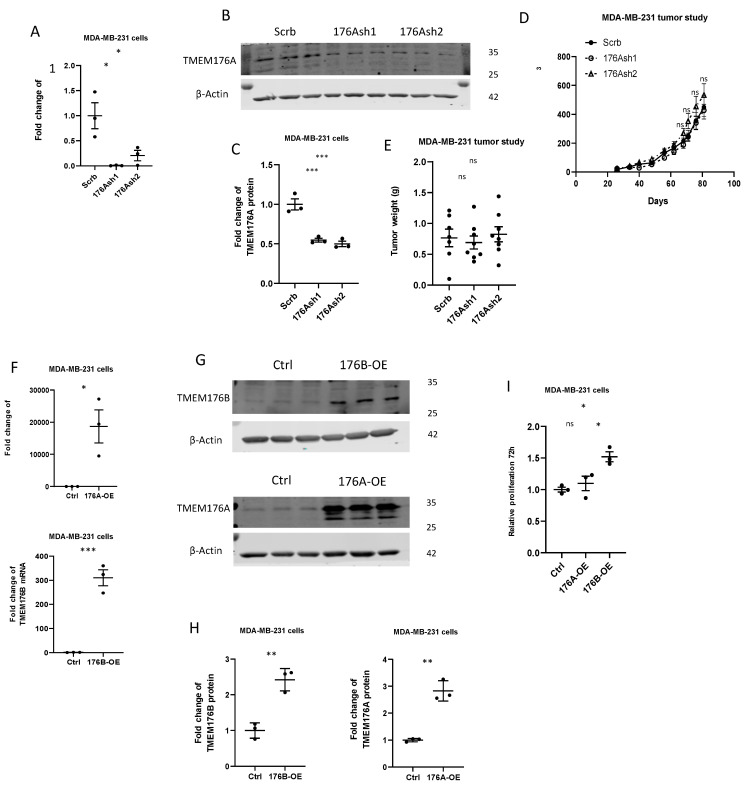
TMEM176B but not TMEM176A affected cell proliferation and tumor growth. (**A**) *TMEM176A* mRNA expression was evaluated by qRT-PCR in control (Scrb) and TMEM176A-silenced (176Ash1, 176Ash2) MDA-MB-231 cells (n = 3 per group, with two independent experiments). (**B**,**C**) Representative Western blot image and quantification of TMEM176A protein expression (n = 3 per group, with three independent experiments). (**D**) Growth charts of MDA-MB-231 control (Scrb), and TMEM176A-silenced (176Ash1 and 176Ash2) tumor xenografts in the Rag1^−/−^ female mice (n = 6–8 mice per group). (**E**) Tumor weight at the end of the study. (**F**) *TMEM176A* and *TMEM176B* mRNA expression was evaluated by qRT-PCR in control (Ctrl) and TMEM176A-overexpressing (176A-OE) or TMEM176B-overexpressing (176B-OE) (n = 3 per group, with two independent experiments). (**G**,**H**) Representative Western blot image and quantification of TMEM176A and TMEM176B protein expression (n = 3 per group, with three independent experiments). (**I**) Proliferation assay of control (Ctrl) and TMEM176A- and TMEM176B-overexpressing MDA-MB-231 cells after 96 h (n = 3 per group in one independent experiment). Data are presented as means ± SEM. Differences between groups were evaluated by Student’s *t*-test (**F**,**H**) and the one-way (**A**,**C**,**E**,**D**,**I**) ANOVA test with the Bonferroni post hoc test. * *p* < 0.05. ** *p* < 0.01. *** *p* < 0.001. ns: not significant.

**Figure 4 cells-10-03430-f004:**
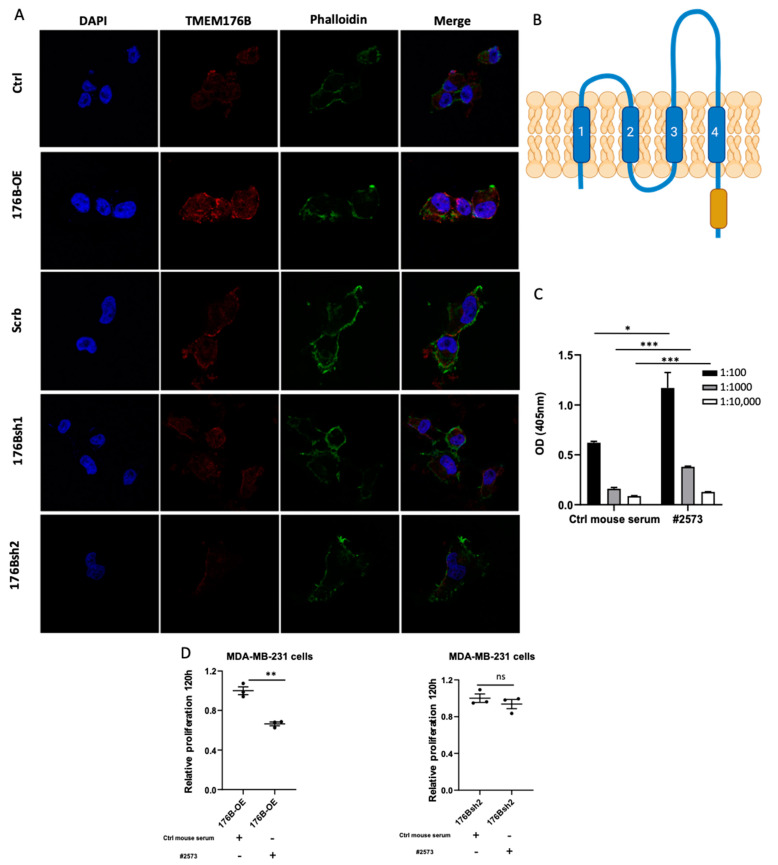
Anti-TMEM176B antibody reduced the proliferation of MDA-MB-231 cells. (**A**) Representative confocal microscopy images of MDA-MB-231 control (Ctrl), TMEM176B-overexpressing (176B-OE), shRNA control (Scrb), and TMEM176B-silenced (176Bsh1, 176Bsh2) cells, stained with TMEM176B and phalloidin (Green) and DAPI nuclear stain (blue). Scale bars, 10 μm. (**B**) Schematic of TMEM176B transmembrane protein with the large extracellular loop 2, which is larger and more diverse (dashed region) than extracellular loop 1 created with BioRender.com (adapted from reference [2]). (**C**) Quantification of the ELISA assay using the TMEM176B polyclonal antibody targeting the large extracellular loop of TMEM176B (n = 3 per group). (**D**) Proliferation assay of TMEM176B-overexpressing cells and TMEM176B-silenced cells treated with normal mouse serum (control) or anti-TMEM176B pAb-2573 (#2573) at 120 h (n = 3 per group, with two independent experiments). Data are presented as means ± SEM. Differences between groups were evaluated by Student’s *t*-test (**C**,**D**). * *p* < 0.05, ** *p* < 0.01, *** *p* < 0.001. ns: not significant.

**Figure 5 cells-10-03430-f005:**
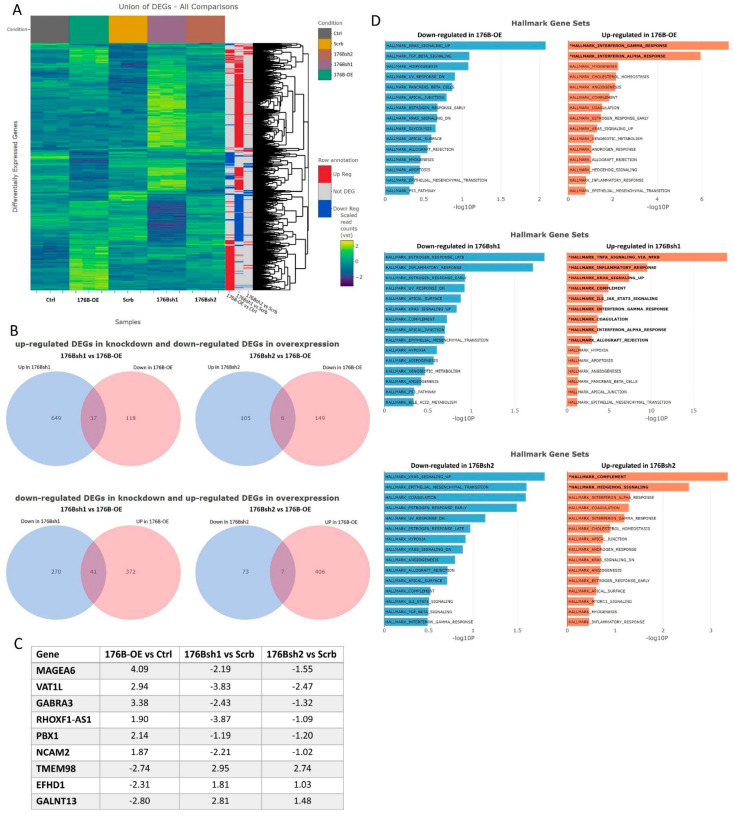
RNA sequencing gene expression analysis in MDA-MB-231 cell lines overexpressing TMEM176B or with TMEM176B gene silencing. (**A**) Heatmaps of all differentially regulated genes (DEGs) between control (Ctrl) and TMEM176B-overexpressing (176B-OE), and shRNA control (Scrb) and TMEM176B-silenced (176Bsh1, 176Bsh2) cell lines to give an overview of the transcriptomic changes. The colored side bars of the heatmap indicate whether the gene is differentially expressed between groups as comparisons stated beneath the colored bars. (**B**) Venn diagrams showing the overlap between the DEGs identified for the cell lines with TMEM176B-silenced and reciprocally expressed in the TMEM176B-overexpressing cells, relative to the respective controls. (**C**) Genes that were identified as significantly differentially expressed in opposite directions between TMEM176B-overexpression and TMEM176B-silenced cell lines. Reported values are the log2 fold changes for their respective controls. (**D**) Functional enrichment via over-representation analysis between control and TMEM176B overexpressing cells, as well as shRNA control (Scrb) and TMEM176B-silenced cells in hallmark gene sets. Gene sets reported in bold with an asterisk are significantly enriched at adjusted *p*-value < 0.05.

**Figure 6 cells-10-03430-f006:**
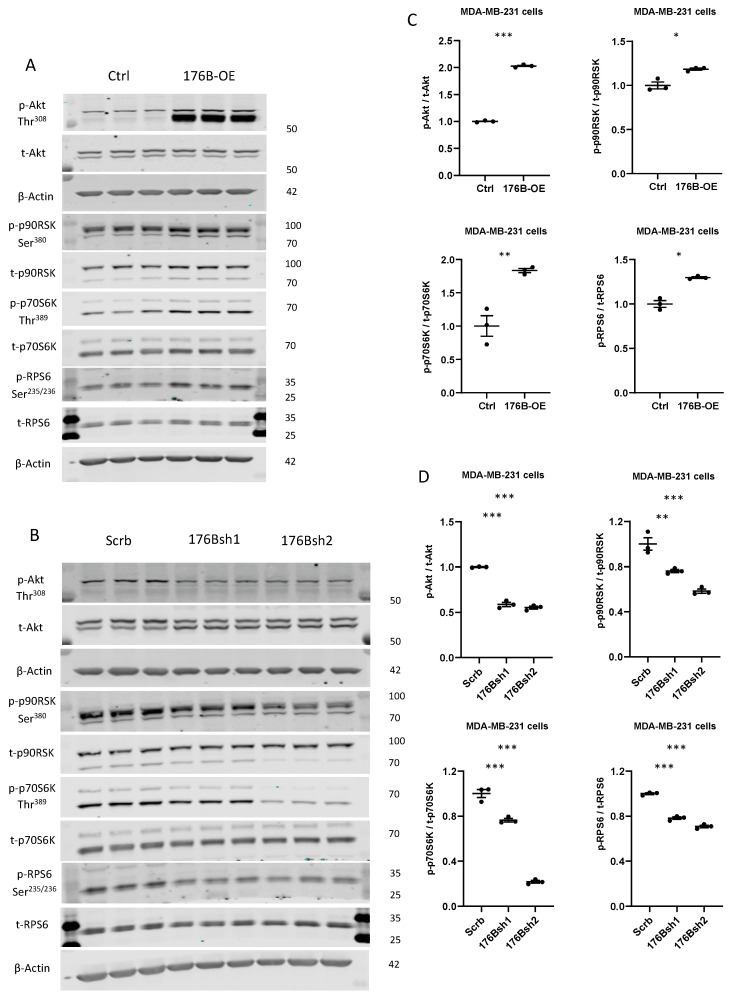
TMEM176B overexpression increased AKT/mTOR pathway activation, which was decreased in TMEM176B-silenced cells. (**A**) Representative Western blot of protein lysates from control (Ctrl) and TMEM176B-overexpressing (176B-OE) cells and (**B**) shRNA control (Scrb) and TMEM176B-silenced (176Bsh1, 176Bsh2) cells examining phospho-AKT (Thr^308^) and total AKT, phospho-p90RSK (Ser^380^), total- p90RSK, phospho-p70S6K (Thr^389^), total-p70S6K, phospho-RPS6 (Ser^235/236^), and total-RPS6 and β-Actin. (**C**,**D**) Densitometry analysis of Western blot was performed using the ImageJ software (n = 3 per group, with three independent experiments). Data are presented as means ± SEM. Differences between groups were evaluated by Student’s *t*-test (**C**) and the one-way (**D**) ANOVA test with the Bonferroni post hoc test. * *p* < 0.05, ** *p* < 0.01, *** *p* < 0.001. 70-kDa ribosomal protein S6 kinase (p70S6K) and ribosomal protein S6 (RPS6) (**A**,**C**). Conversely, in the TMEM176B-silenced cells, we found reductions in AKT (Thr^308^) p90RSK, p70S6K, and RPS6 phosphorylation compared with control cells (**B**,**D**). These results suggest that the cation channel TMEM176B regulates the AKT/mTOR signaling pathway.

## Data Availability

Gene expression data will be publicly available through submission to Gene Expression Omnibus (GEO) upon acceptance of the manuscript.

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
