# Peer review of "TMEM176B Regulates AKT/mTOR Signaling and Tumor Growth in Triple-Negative Breast Cancer"

_cells, 2021, doi:10.3390/cells10123430_

Round 1

Reviewer 1 Report

The manuscript by Kang et al. entitled ″TMEM176B regulates AKT / mTOR signaling and tumor growth in triple negative breast cancer″ is very interesting and provides new insights and perspectives regarding TNBC. I believe that the manuscript is well-written and comprehensive, and the methods used in the study are appropriate and modern. However, there are a few detailes that have to be fixed before publishing the manuscript:

-starting from the abstract, and throughout the manuscript, TMEM176B is written differently many times (in Uppercase, sentence case, cursive, italic etc). I think it should be written in the same way everywhere to confer uniformity to the manuscript

-the same is applible to AKT (it appears as Akt on 357,360,367)

-please correct c-myc/vegfa from line 60 (c-Myc/VEGFA)

Other than that, I think that this is a very good manuscript.

Author Response

Thank you to the reviewer for these comments. 

To address each comment specifically:

1. starting from the abstract, and throughout the manuscript, TMEM176B is written differently many times (in Uppercase, sentence case, cursive, italic etc). I think it should be written in the same way everywhere to confer uniformity to the manuscript

- We initially used the different styles to indicate mouse and human genes and proteins. We have cleaned up the styles so when discussing TMEM176B expression at a protein level everything is capitalized, it is italicized for human genes and the lower case us only used when specifically referring too mouse RNA expression. 

2. the same is applible to AKT (it appears as Akt on 357,360,367)

- Thank you. this was an oversight and we have changed the lower case Akt to AKT as it all refers to protein expression

3. please correct c-myc/vegfa from line 60 (c-Myc/VEGFA)

- We have corrected this too. initially it was lower case as it was referring to mouse genes, but it the mice also overexpress the protein, so it makes sense to change it. 

Reviewer 2 Report

In this study, Kang et al. report that TMEM176B regulates AKT / mTOR signaling pathway and contributes to breast cancer growth and progression. Over the past years, TMEM176B has been associated with cancer pathology. This study further points out the TMEM176B also plays a role in breast cancer. The demonstration of these results in cancer cells lacks novelty. However, the authors use multiple methods to show the critical role of TMEM176B in breast cancer growth by direct effects on tumor cell signaling, gene expression, tumor microenvironment, and immune response. In my view, the manuscript is complete and provides reliable data and controls in its current form. While this study has clear merits, it largely lacks relevance to organelle function or “Organellar Ion Channels and Transport Proteins in Health and Disease,” which is a significant limitation.

Author Response

While this study has clear merits, it largely lacks relevance to organelle function or “Organellar Ion Channels and Transport Proteins in Health and Disease,” which is a significant limitation.

  • We thank the reviewer for this clearly important comment and we agree that we did not include enough information to make it apparent why we felt our manuscript fit into this theme. We have now added some additional references and context regarding what is known about TMEM176B from previous studies. Much of the work on TMEM176B has been done by Cedric Louvet's group in Nantes. They. have found it to be an ion channel on endo-phagosomes that regulates endosomal pH, and have tied it through this mechanism to dendritic cell function.
  • We have updated some parts of the text too highlight the important links between ion channels in the regulation of endosomal pH, and in turn cell signaling. Although we don't have all of the answers at this point as to exactly how TMEM176B regulates cell signaling. We hope this provides better context for why we felt the manuscript fit with the theme of these articles. Thank you again for this point, we appreciate the opportunity to address it and improve the manuscript.